# Neuro-Inflammaging and Psychopathological Distress

**DOI:** 10.3390/biomedicines10092133

**Published:** 2022-08-31

**Authors:** Giuseppe Murdaca, Francesca Paladin, Marco Casciaro, Carmelo Mario Vicario, Sebastiano Gangemi, Gabriella Martino

**Affiliations:** 1Department of Internal Medicine, University of Genoa, 16132 Genoa, Italy; 2Ospedale Policlinico San Martino IRCCS, 16132 Genoa, Italy; 3Department of Biomedical and Dental Science and Morphofunctional Imaging, University of Messina, 98125 Messina, Italy; 4COSPECS Department, University of Messina, 98122 Messina, Italy; 5Allergy and Clinical Immunology Unit, Department of Clinical and Experimental Medicine, University of Messina, 98125 Messina, Italy; 6Department of Clinical and Experimental Medicine, University of Messina, 98122 Messina, Italy

**Keywords:** chronic inflammatory diseases, inflammaging, pro-inflammatory cytokines, neuroinflammation, clinical psychology, psychopathological disorders

## Abstract

Inflammaging is a low degree of chronic and systemic tissue inflammation associated with aging, and is intimately linked to pro-inflammatory mediators. These substances are involved in the pathogenesis of chronic inflammatory diseases and related psychopathological symptoms. When inflammation and aging affect the brain, we use the term neuro-inflammaging. In this review, we focused on the neuro-inflammatory process typical of advanced ages and the related psychopathological symptoms, with particular attention to understanding the immune-pathogenetic mechanisms involved and the potential use of immunomodulatory drugs in the control of clinical psychological signs. Inflammation and CNS were demonstrated being intimately linked in the neuro-inflammatory loop. IL-1, IL-6, TNF-a, COX and PGE are only partially responsible. BBB permeability and the consequent oxidative stress resulting from tissue damage make the rest. Some authors elaborated the “theory of cytokine-induced depression”. Inflammation has a crucial role in the onset symptoms of psychopathological diseases as it is capable of altering the metabolism of biogenic monoamines involved in their pathogenesis. In recent years, NSAIDs as an adjunct therapy in the treatment of relevant psychopathological disorders associated with chronic inflammatory conditions demonstrated their efficacy. Additionally, novel molecules have been studied, such as adalimumab, infliximab, and etanercept showing antidepressant and anxiolytic promising results. However, we are only at the beginning of a new era characterized by the use of biological drugs for the treatment of inflammatory and autoimmune diseases, and this paper aims to stimulate future studies in such a direction.

## 1. Introduction

In recent decades there has been a progressive increase in the prevalence of chronic diseases, mainly due to an ever-increasing life expectancy. Aging and chronic diseases find their fil rouge in chronic inflammation.

Aging and inflammation have been defined in their interplay since the 1991 New York Academy of Sciences conference by a group of researchers [1]. Among these, Claudio Franceschi conceptualized the word “inflammaging”. He and his group elaborated that a constantly growing survival, together with the aging process affected by a biological, chemical, and physics damage led to a chronic inflammatory process. Longevity is a characteristic of modern industrialized society, and successful aging is the key to countering destructive processes due to several causes and contributing to organ damage. Franceschi et al. also speculated a series of defensive systems against chronic inflammatory injuries in the elderly, called “anti-inflammaging”. They evaluated centenarians who were able to escape the diseases typical of advanced ages. These kinds of subjects were defined as “escapers” [2,3].

Although different tissues and organs are usually targeted by long-term inflammation, one of the most sensitive due to its limited renewal capacity is the brain. Neurodegenerative diseases are a direct consequence. The link between the molecular and cellular balance capable of permitting a physiological healthy aging or a cognitive impairment is still unclear.

The actual models suggest that exposure during lifespan to several exogenous and endogenous insults triggers an immune response, inducing a state of chronic physiological inflammation that is protective for long-term survival at certain levels. As mentioned above, a systemic inflammatory state defines a phenotype typical of successful aging. Moreover, neurodegeneration is affected by an age-related increase in inflammation, with simultaneous adaptive activation of anti-inflammation procedures [4].

The innate immune system plays a crucial role in the inflammatory processes, usually reduced at advanced ages. Additionally, hyperreactivity could also be age-associated. Microglia are main actors among the resident immune cells in the brain. In the elderly, senescent microglia augment the production of proinflammatory mediators with reduced chemotaxis and phagocytosis capacities, particularly of amyloid-β fibrils [5]. Geriatric microglia amass mitochondrial DNA with detrimental effects on cells leading to ROS accumulation and sustaining further damages.

Also, the reduced effectiveness of the adaptive immune system contributes to declined immune response versus biological threats [6].

The raised systemic inflammatory state and peripheral immunosenescence interfere with neuronal immune cell activity and reactivity. The consequence is a chronic low-grade inflammatory condition called neuro-inflammaging. The activated glia in a loop sustained by cytokines, oxidative stress, and damage is mainly involved in memory loss and cognitive decline. Immunosenescence and inflammaging induce brain suffering, with the subsequent spread of clinical psychological signs and symptoms such as cognitive impairment, behavior alteration, or mental disorders [7].

This pervasive process of aging, known as “inflammaging”, is characterized by a low degree of chronic and systemic tissue inflammation, revealed by high levels of some biomarkers such as C-reactive protein (CRP), interleukin-6 (IL-6), and tumor necrosis factor-alpha (TNFa) [1]. Growing evidence has drawn attention to the potential role of these and other inflammatory mediators in the pathogenesis of chronic inflammatory diseases and related psychopathological symptoms, such as depression, anxiety, and alexithymia as pervasive and worsening symptoms and signs of the organic pathology. Chronic neuroinflammation plays a primary pathophysiological role in developing neurotoxic alterations of those brain regions involved in emotional regulation, contributing i.e., to the development of major depressive disorder (MDD) [8].

This review is based on the role of this neuro-inflammatory process in the correlation between chronic inflammatory diseases typical of advanced ages (neuro-inflammaging) and psychopathological symptoms, with particular attention to understanding the immune-pathogenetic mechanisms involved and the potential use of immunomodulatory drugs in the management of clinical psychological signs.

## 2. Epigenetics, Aging, and Chronic Inflammation

Chronic inflammatory diseases are characterized by a condition of systemic or organ-specific inflammation which causes tissue damage, i.e., a process of cell death [9].

Studies conducted so far suggest that epigenetic changes play a fundamental role in regulating the pathophysiological processes that lead to the development of chronic inflammatory diseases such as diabetes, cardiovascular disease, cancer, and neurological disorders. These epigenetic modifications such as DNA methylation, post-translational modifications to histone proteins, and non-coding RNA expression can be hereditary or environmentally induced [10]. Among the environmental factors favoring the onset of epigenetic changes and therefore the development of chronic inflammatory diseases we remember obesity and unhealthy diet, as well as bad lifestyle habits including sleep deprivation, chemical exposure, alcohol abuse, smoking, and climate pollution [11].

Additionally, aging represents one of the factors favoring the development of epigenetic defects, making the elderly the category most prone to the development of age-related chronic diseases [12]. Chronic inflammation is indeed a pervasive feature of aging [13]. This low-grade inflammatory condition typical of old age is referred to as “inflammaging” [14]. This condition can occur due to an augmented concentration of proinflammatory mediators, or due to a deficient anti-inflammatory response.

### 2.1. Proinflammatory Mediators

Interleukin-6 (IL-6), tumor necrosis factor alpha (TNFα) and C-reactive protein (CRP) are among the pro-inflammatory mediators most involved in the inflammaging process, as well as several mediators secreted by the monocyte/macrophages such as TNFα and Interleukin-1 (IL-1), as well as chemokines such as MCP-1 and Interleukin-8 (IL-8) [15].

IL-1 is one of the main inflammatory cytokines with a pyrogen function. It is generated and released by several cells such as macrophages, monocytes, fibroblasts, and dendritic cells (DCs), B lymphocytes, natural killer (NK) cells and epithelial cells. It boosts cell proliferation, differentiation, and promotes innate and specific immune responses. IL-1 acts as the main actors in various inflammatory-mediated diseases triggering the immune system [16].

IL-6 is a cytokine with pleiotropic activities with multiple functions. It regulates immune responses, hematopoiesis, acute phase responses, having a central role in inflammation. Due to the context, it can exert pro- and anti-inflammatory actions with various signaling paths. IL-6 is provided by endothelial cells, fibroblasts, monocytes, and macrophages in response to diverse triggers such as IL-1, IL-17, and TNF-a during the inflammatory response [16]. Higher plasma concentrations of inflammatory factors such as IL-6 and TNF-α have been identified as predictors of mortality in the elderly, demonstrating the interconnection between their immune and functional status [17]. IL-6 has been shown to be associated, for example, with a dysregulation of lipid metabolism and worse cardiac mitochondrial oxygen uptake and therefore an increase in the incidence of cardiovascular diseases [18].

Tumor necrosis factor was firstly isolated as a circulating molecule capable of provoking necrosis in tumors; secondly it was better characterized as a main actor in inflammation. TNF can act on diverse receptors, generating different signaling pathways inducing cell death, survival, differentiation, proliferation and migration. It takes part in the inflammaging process, together with IL-1b and IL-6 [15,19]. TNF-α is one of the main pro-inflammatory molecules involved in neurological disorders and current data suggest that its down-regulation could also improve cognitive function in relation to its ability to cross the intact blood brain barrier (BBB) involving the type I and II TNF-α receptors (TNF-RI and TNF-RII). TNF-α is in fact expressed at physiological levels by microglia and neurons; increasing its expression in activated microglia, neurons, oligodendrocytes, reactive astrocytes, epithelial cells, endothelial cells and ependymal cells on brain and peripheral lesions and in chronic disorders [20].

Additionally, IL-18 can be listed among the proinflammatory cytokines. It is capable of triggering lymphocytes as a potent defense against severe infections. IL-18, depending on the pleiotropic environment can induce both Th1 and Th2 responses [21,22].

Regarding the role of CRP in chronic inflammatory diseases, growing evidence suggests that elevated CRP concentrations are associated with an increased risk of cardiovascular diseases (CVD), type 2 diabetes mellitus (T2DM), Alzheimer’s disease (AD), hemorrhagic stroke and Parkinson’s disease (PD); thus representing not only an excellent biomarker of chronic inflammation, but also a direct protagonist of its pathological process [23,24].

In particular, IL-1α and IL-1β known as IL-1 and IL-18, represent the two molecules responsible for the initiation of the inflammatory cascade induced by stress. Studies conducted on the elderly population, including centenarians, have shown that the increase in these mediators is closely related to an increase in comorbidity, age-related disease and mortality [25]. However, IL-18 is also endowed with the ability to stimulate innate immunity, thus improving the host’s defense against infections. For this reason, it is hypothesized that since large elderly subjects have increased levels of this biomarker, it may under certain circumstances be more relevant for survival than the negative potential mainly on the vascular system [26]. Experimental studies conducted on centenarian subjects have shown how these presented increased levels of IL-18 and its natural antagonist, the high affinity binding protein (IL-18BP), compared with the control cohort. These findings, therefore, demonstrate the crucial role of the immune-inflammatory response in the anti-aging process [27].

The inflammatory status could also be sustained by an inappropriate localization of DNA within the cell cytosol. This condition which can occur when a cell may have a problem with its own nuclear integrity is “sensed” by some specific receptors that trigger cytokine production and give start to an inflammatory and lytic cell disruption known as pyroptosis. DNA sensing is a fundamental process in physiological conditions, but when it is unbalanced it may favor autoimmune diseases and premature aging [28,29]. For example, some proteins such as AIM2 are able to sense DNA. They are able to assemble multiprotein complex called inflammasome and start a pro-inflammatory response leading to pyropitosis and the cleavage of pro-inflammatory cytokines such as pro-IL-1β and pro-IL-18 favoring inflammation [30,31].

### 2.2. The Anti-Inflammatory Counterbalance

On the other hand, inflammaging seems to be determined not only by an increase in some pro-inflammatory cytokines but also by an age-related reduction in those with anti-inflammatory action, such as interleukin-10 (IL-10).

IL-10 is an anti-inflammatory cytokine of exceptional power. It can be released by both innate and adaptive immune cells. IL-10 exerts its potent immunosuppressive action by blocking the inflammatory process at diverse stages [32]. It acts directly and indirectly on both the innate and adaptive paths, by inhibiting the production of proinflammatory cytokines, antigen presentation, and cell proliferation [15]. For example, IL-10 is responsible for the suppression of the proinflammatory response in various tissues, which it exercises by blocking the release of inflammatory cytokines, such as IL-6, TNF-α, and IL- 1β [33,34].

Moreover, the role in immunosenescence acted by cytokine receptors and the consequent signalling should not be underrated. Of course, evaluating receptors’ role is more arduous as several cytokine receptors are common to a family of cytokines. In this context, often their specificity is assured by the individual subunits. The pathway influenced by the receptor could favor or not favor inflammation [35].

## 3. Cytokines and Neuroinflammation

Studies conducted over the years have shown that cytokines are important in brain development, being able to promote healthy brain function by supporting neuronal integrity, neurogenesis and synaptic remodeling. It has also been widely demonstrated that cytokines also have the ability to influence neurocircuits and neurotransmitter systems, causing behavioral alterations such as depression and anxiety [36].

The central nervous system (CNS) appears to be particularly vulnerable in dysregulated cytokine networks. The local neuronal tissue response to leukocyte invasion is characterized by the production of cytokines such as IL-1, IL-6 and TNF-α. These cytokines are able to act on the cells resident in the CNS to stimulate the production of a more diverse range of cytokines, thus fueling an inflammation waterfall [37].

Peripheral cytokines are able to access the central nervous system, where they increase the production of local inflammatory mediators such as cyclooxygenase-2 (COX-2), prostaglandin E2 (PGE2), nitric oxide (NO), cytokines and chemokines by endothelial cells, perivascular macrophages and microglia. The result is a progressive and chronic increase in oxidative stress favored by the generation of reactive oxygen and nitrogen species (ROS and RNS), which lead to the oxidation of some cofactors necessary for the synthesis of monoamines. Furthermore, there is evidence indicating that inflammatory cytokines and their signaling pathways may decrease the expression or function of the vesicular monoamine transporter 2 (VMAT2) and/or increase the expression or function of serotonin and transporters. dopamine (5-HTT/DAT). Cytokines can also decrease brain-derived neurotrophic factor (BDNF), negatively affecting neurogenesis and neuroplasticity. Finally, inflammatory cytokines can affect the glutamate (Glu) system by activating the enzyme, indoleamine-2,3-dioxygenase (IDO), which catabolizes tryptophan, the primary amino acid precursor of 5-HT in kynurenine, contributing to excitotoxicity [38,39,40].

The abnormal activation of microglia resulting from this neuro-inflammatory process is involved in the pathogenesis of numerous neurodegenerative diseases including AD and PD, but also of psychiatric disorders such as stress, depression and schizophrenia [41]. The acute form of neuroinflammation is associated with glia activation of the central nervous system (CNS) with abnormal increases in cytokines and chemokines, infiltration of peripheral immune cells, edema, increased permeability of the blood brain barrier (BBB) and disruption. This degree of neuroinflammation is associated with autoimmune diseases such as multiple sclerosis (MS), where it is responsible for the demyelination process of axons, up to the chronicization of neuroinflammation and the consequent irreversible loss of axons [42].

The chronicization of neuroinflammation results in strong changes in the inflammatory profile of microglia, as a consequence of a continuous release of neurotoxic inflammatory mediators, such as inducible nitric oxide synthase (iNOS) and pro-inflammatory cytokines, including IL-1 and tumor necrosis factor TNF-α, clinically resulting in an increased predisposition to the development of cognitive impairment and therefore also associated mood disorders, such as depression [43].

Chronic neuro-inflammation is typical of aging, and the brain is particularly sensitive to damage. Cognitive impairment is a consequence that affects morbidity and mortality in the elderly population. Another main role in the inflammatory response in the aged brain is led by the inflammasome. The inflammasome is a multi-protein complex which is an essential part of the innate immune response. They can sense the danger signals, integrate them in the cell and respond by producing specific pro-inflammatory cytokines. The inflammasome is a key player in activating caspase-1 and stimulating the inflammatory cytokines interleukin IL-1β and IL-18 [44]. The trigger of inflammasomes starts when immune cells face pathogen-associated molecular patterns (PAMPs) or danger-associated molecular patterns (DAMPs) by pattern recognition receptors (PRRs) [45]. During recent years, it was demonstrated that NLRC4, caspase-1, ASC and IL-18 are elevated in the aged brain, confirming its key role in the innate immune response [46,47].

## 4. Correlation between Depression and Other Psychopathological Disorders in Chronic Inflammatory Diseases

According to Maes’ “theory of cytokine-induced depression” [41], inflammatory factors play a crucial role in the onset symptoms of psychopathological diseases as they are capable of altering the metabolism of biogenic monoamines involved in their pathogenesis, i.e., dopamine, noradrenaline and serotonin in the mesencephalic nuclei [48,49].

It follows that both depression and some chronic inflammatory diseases such as cardiovascular diseases, respiratory diseases, metabolic disorders and autoimmune diseases have the same immunological background. These chronic diseases represent not only the risk factors for a depressive episode, but also factors of resistance to psychoactive therapies and favoring relapses of episodes of deteriorated mood [50,51]. Recent studies have in fact highlighted how the use of mood stabilizers was associated with lower IL-6 levels at the follow-up of patients with major depressive disorder [52].

Subsequent studies have shown associations between cytokine production and the presence of both depressive and anxious symptoms, in relation to adjustments for favoring factors such as smoking and alcohol intake. Specifically, lipopolysaccharide-stimulated inflammation (LPS) was more consistently associated with anxiety levels in which the presence of pro-inflammatory factors such as IL-6, IL-8, IL-10, IL-18, MCP-1, MMP2 and TNF-β prevails, with respect to depressive symptoms, in which molecules such as IL-8, MCP-1 and MMP2 prevail [53,54,55,56].

Similarly, patients with elevated CRP levels appear to be associated with greater symptom severity, a specific pattern of depressive symptoms, and worse response to treatment. Additionally, approximately one-third of depressed patients exhibited low-grade inflammatory status (i.e., CRP > 3.0 mg/L), suggesting the presence of a different subgroup of Major Depressive Disorder (MDD) with distinct etiopathogenesis, and prognosis, which could potentially respond to anti-inflammatory treatments [57,58].

Depressive disorders are predominantly associated with some chronic inflammatory/autoimmune diseases. This is what happens, for example, in rheumatoid arthritis, where depression represents an important co-morbidity which is more represented, compared to the general population [59].

Even in patients with chronic obstructive pulmonary disease (COPD) with depression in comorbidities there was an increase in the concentration of inflammatory markers such as IL-1 and TNF-α salivary and a decrease in salivary cortisol. In particular, IL-1 and TNF-α showed a significant positive correlation with the severity of depression, while salivary cortisol showed a negative correlation with the severity of depression in the COPD patients [60].

Another relevant factor that has been shown to negatively influence the extent of depressive symptoms and which seems to share the inflammatory pattern and the correlation with various chronic inflammatory diseases, is represented by alexithymia, the difficulty in identifying, recognizing, distinguish, and express the own and others feelings [61,62,63,64]. Similarly, with specific regard to alexithymia, recent studies highlighted a pathogenetic correlation with specific inflammatory markers such as TNF-α and IL-6 exists [36,37,38,65,66,67]. Finally, it should be mentioned the literature reporting the negative influence of chronotype on depressive symptoms [68], which also significantly predicts the risk to develop anxiety disorders [69] and inflammatory diseases, such as Irritable Bowel Syndrome [70]. All these studies have associated late chronotype with a higher risk for the above-mentioned clinical conditions.

## 5. Criticisms: Altered Cytokine Patterns

Inflammation and CNS were demonstrated being intimately linked in the neuro-inflammatory loop. Most of the damages and alteration conducting to an altered physiology are due to an impaired cytokine generation and permeability. IL-1, IL-6, TNF-α, COX and PGE are only partially responsible. BBB permeability and the consequent oxidative stress resulting from tissue damage make the rest leading to axons loss. Not only cells, but also mediator concentration gives a clinically significant scenario; dopamine, noradrenalin and serotonin alteration lead to the development of psychopathological symptoms. These indicators are often present in neurodegenerative diseases which have lot in common, such as AD and PD. Often inflammation travel next autoimmunity and also MS has a similar clinical development. Anxiety, depression and impaired mood are often present also in autoimmune diseases where, as mentioned above, very frequently pro-inflammatory mediators are high. The main quest is balancing the altered cytokine pattern (Figure 1).

From our literature analysis, it emerged that the neuro-inflammation process and in particular some pro-inflammatory mediators such as CRS, Il-6 and TNF- α, represent the link between some chronic diseases (CVD, T2DM, AD and arteriosclerosis) and psychopathological disorders and signs, with particular reference to DCS and alexithymia. This immuno-pathogenetic correlation, known as Maes’ “theory of cytokine-induced depression” [41], is determined by the now known ability of some pro-inflammatory cytokines to cross the blood-brain barrier (BBB), altering the processes of neuronal plasticity and release of neurotransmitters involved in the genesis of some psychiatric disorders. Clinical studies conducted on humans have in fact shown that subjects affected by DCS have higher levels of circulating cytokines than controls [49,71], hence the potential role of these pro-inflammatory mediators as predictors of the therapeutic response, with a negative correlation, i.e., higher levels of pro-inflammatory cytokines would lead to a poor response to antidepressant treatment [72,73].

These pro-inflammatory cytokines are also involved in the induction of some chemokines such as fractalkine or FKN, also called chemokine ligand 1 with C-X3-C motif; CX3CL1) of the CX3C subfamily, which has been shown to be involved in various associated non-immune mechanisms to clinical psychological disturbances. This is in fact able to alter the neurotransmission processes by inhibiting the serotoninergic one by enhancing the activity of GABA, and the glutamatergic one in the hippocampus region, thus contributing to the genesis of depressive-anxious symptoms, suggesting how FKN can represent a target for a specific immune therapy for this disease [74,75].

## 6. Future Prospective

For the reasons explained above, in recent years, attention has focused on the introduction of some anti-inflammatory drugs as adjunct therapy in the treatment of relevant psychopathological disorders associated with chronic inflammatory conditions [76,77,78]. Among the adjuvant drugs that have been studied we find, for example, non-steroidal anti-inflammatory drugs (NSAIDs) such as celecoxib [74,75], which has been shown to be able to mainly reduce IL-6 values by improving the extent of depressive symptoms, although it does not represent a viable long-term treatment in relation to the development of side effects, mainly in subjects over 65 years of age. In addition to NSAIDs, other molecules have also been studied, such as the anti-TNF-α monoclonal antibodies (adalimumab, infliximab and etanercept), in which they have demonstrated the ability to exert antidepressant and anxiolytic effects when administered to patients with inflammatory diseases [79,80]. Despite these promising results, we are only at the beginning of a new era characterized by the use of biological drugs for the treatment of several inflammatory and autoimmune diseases. Their effectiveness in reducing cytokines associated with mood disorders could be of extreme importance. Multiple drugs association, their dosage and timing of administration are the next challenge in order to be a game changer in several psychopathological impairment. For this reason, this paper aims to stimulate future studies in such a direction since cognitive disorders constitute a heavy burden for lot of patients.

## Figures and Tables

**Figure 1 biomedicines-10-02133-f001:**
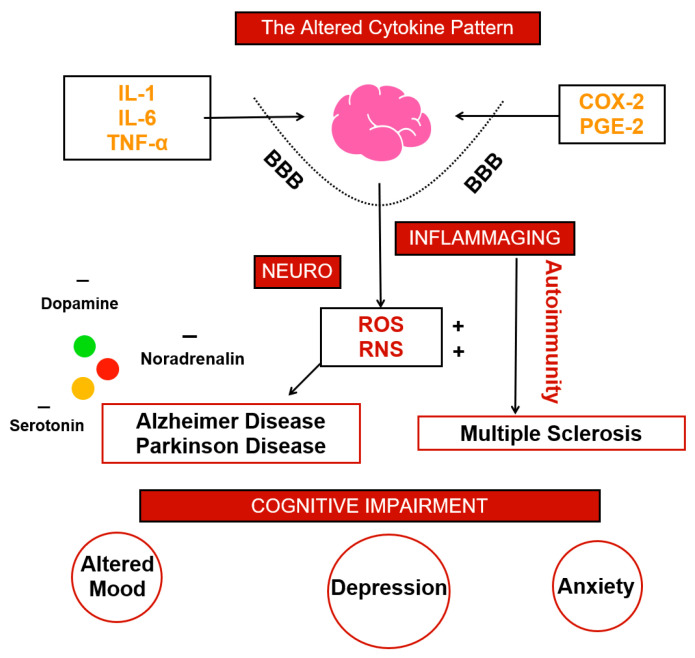
In Figure 1 we summarize the altered cytokine pattern at the basis of brain inflammation which in turn leads to cognitive impairment and mood modification.

## Data Availability

Not applicable.

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
