# Peer review of "Neuro-Inflammaging and Psychopathological Distress"

_biomedicines, 2022, doi:10.3390/biomedicines10092133_

Round 1

Reviewer 1 Report

The authors present an interesting review on the concept of ‘inflammaging’ – a relatively new term which defines the influence of ageing of the physiological processes of the body which may lead to the initiation and development of chronic disorders. Within, the authors highlight some of the more pertinent cytokines and inflammatory markers linked to these pathways, while also focussing on these pathways in the context of neurological disorders such as Alzheimer’s and similar. In this way, the therapeutic efficacy of anti-inflammatories in this context are also discussed.

In reviewing the manuscript however I had a number of concerns. The following should be addressed when preparing a suitable revision.

1.       While the writing is good for the most part, there are a few instances where non-scientific language or terminology is used.  In this way, the language of the piece should be reviewed in the interest of bringing the article closer to publication standard.

2.       The structure of the review could be improved upon. Under certain sections, several cytokines are discussed individually. It would be an idea to use subsections to discuss each of these cytokines in their own right to improve the flow of information within the article.

3.       Similar to above, the lack of structure also sees certain elements overdeveloped when compared to others. For example, the information on IL-10 feels tacked onto the end of Section 2 without any real development of that cytokine with respect to the review. The balance and flow of the review needs attention in this way.

4.       While a great deal of focus is placed upon the profile of cytokines in the context fo inflammaging, it might be an idea to include some information on the effect of ageing on cytokine receptor expression.   

5.       Certain sections are quite light on referencing.

Author Response

See attached the cover letter with point-to-point resppnse to the reviewers

Giuseppe Murdaca

Reviewer 2 Report

Neuro-inflammaging has been very hot field and well-addressed in many other reviews.  This submitted manuscript tried to link neuro-inflammaging with psychopathological diseases which is a good direction; however, the summarization is not enough and dose not give the readers a whole picture.   For example, the abstract is not a good summarization for this manuscript and does not show close relationship with the title.  The subtitles are not suitable and meaningful, such as “Pro-inflammatory markers involved in chronic diseases” and “Cytokines and neuroinflammation”.  The different parts of this manuscript have no inherent linkage and are very difficult to follow.  In general, this manuscript did not provide a full scope of neuro-inflammaging and diseases and did not present insightful comments. 

In addition, there are some sentences which are difficult to follow and need to rephrase.  There is even a typo in the title which raises much concerns on the whole quality of this manuscript. 

Author Response

See cover letter with point-to-point response to reviewers

Giuseppe Murdaca

Reviewer 3 Report

The review is interesting, but needs to be revised, for example, citation references should be added to Sections 4 [Correlation between depression and other psychopathological disorders in chronic inflammatory diseases] and 7 [Future prospective], and tables should be added to make it easier for the reader to read.

Author Response

See attached cover letter with point-to-point response to reviewers

Giuseppe Murdaca

Round 2

Reviewer 1 Report

The authors have addressed my comments and the manuscript is much improved. 

Author Response

Please find enclosed the cover letter with the response to reviewer suggestions

Reviewer 2 Report

The authors have addressed some concerns raised by the reviewer.  Still the reviewer has some concerns on the whole structure/organization of this manuscript.   The title indicates they are going to discuss some psychopathological diseases but the context was only on depression.  Actually they have several psychopathological diseases (anxiety, bipolar diseases, PTSD, ect.).   

Author Response

(The authors gave the same response as above.)

Reviewer 3 Report

1 Please refer to the following references in Section 4 [Cytokines and neuroinflammation], paragraph 2

Marc UdinaJosé Moreno-EspañaRicard NavinésDolors GiménezKlaus LangohrMònica GratacòsLucile CapuronRafael de la TorreRicard SolàRocío Martín-Santos. Serotonin and interleukin-6: the role of genetic polymorphisms in IFN-induced neuropsychiatric symptoms. Psychoneuroendocrinology 2013 Sep;38(9):1803-13. doi: 10.1016/j.psyneuen.2013.03.007. Epub 2013 Apr 6.

2 Please check the section number, there are 2 sections 4.

3 Please unify TNF- α Representation (L198,  TNF-α, L238, TNF-alpha,ha-L334, TNFa-a).

Author Response

(The authors gave the same response as above.)

Round 3

Reviewer 2 Report

The authors have addressed the reviewer's concerns